# Enhanced Detection in Droplet Microfluidics by Acoustic Vortex Modulation of Particle Rings and Particle Clusters via Asymmetric Propagation of Surface Acoustic Waves

**DOI:** 10.3390/bios12060399

**Published:** 2022-06-10

**Authors:** Yukai Liu, Miaomiao Ji, Nanxin Yu, Caiqin Zhao, Gang Xue, Wenxiao Fu, Xiaojun Qiao, Yichi Zhang, Xiujian Chou, Wenping Geng

**Affiliations:** 1Science and Technology on Electronic Test and Measurement Laboratory, North University of China, Taiyuan 030051, China; s2006073@st.nuc.edu.cn (Y.L.); s202106120@st.nuc.edu.cn (N.Y.); s1906211@st.nuc.edu.cn (C.Z.); s1906061@st.nuc.edu.cn (G.X.); xiaojunqiao@nuc.edu.cn (X.Q.); 20220039@nuc.edu.cn (Y.Z.); chouxiujian@nuc.edu.cn (X.C.); 2Key Laboratory of Instrumentation Science & Dynamic Measurement, North University of China, Ministry of Education, Taiyuan 030051, China; s2006240@st.nuc.edu.cn; 3School of Mechanical Engineering, North University of China, Taiyuan 030051, China; s2002032@st.nuc.edu.cn

**Keywords:** droplet microfluidics, biochemical analysis, analytical methods, travelling SAW

## Abstract

As a basis for biometric and chemical analysis, issues of how to dilute or concentrate substances such as particles or cells to specific concentrations have long been of interest to researchers. In this study, travelling surface acoustic wave (TSAW)-based devices with three frequencies (99.1, 48.8, 20.4 MHz) have been used to capture the suspended Polystyrene (PS) microspheres of various sizes (5, 20, 40 μm) in sessile droplets, which are controlled by acoustic field-induced fluid vortex (acoustic vortex) and aggregate into clusters or rings with particles. These phenomena can be explained by the interaction of three forces, which are drag force caused by ASF, ARF caused by Leaky-SAW and varying centrifugal force. Eventually, a novel approach of free transition between the particle ring and cluster was approached via modulating the acoustic amplitude of TSAW. By this method, multilayer particles agglomerate with 20 μm wrapped around 40 μm and 20 μm wrapped around 5 μm can be obtained, which provides the possibility to dilute or concentrate the particles to a specific concentration.

## 1. Introduction

Biological detection and diagnosis technology has always been a research hotspot in the medical field. In microscale fluids, if the sample of tiny objects (cells or particles) can be quickly and inexpensively used to capture and enrich, it can significantly reduce the capacity of fluid, and greatly improve the sensitivity and accuracy of sample analysis and detection [1]. Surface Acoustic Wave (SAW)-based devices have the advantages of real-time, label-free detection capabilities, remote control function and simple operation [2]. Samples such as cells and microorganisms can be manipulated without harmful damage from electric or magnetic fields, and without cross-contamination due to direct contact between manipulator and sample [3]. An acoustofluidic miniaturized system composed of piezoelectric substrate and closed microchannel/chamber has been widely used in particle sorting [4], washing [5], enrichment [6], and arrangement [7]. The acoustic radiation force (ARF) generated by surface acoustic waves can accurately manipulate and capture particles or cells suspended in liquid, which has great application potential [8]. However, sample detection requires only one drop or less in most testing cases [9], and also the addition of microchannels, catheters and injection pumps can make the detection system complex and difficult to carry, the difficulty of operation and detection time will also increase at the same time [10]. Another simpler and more ingenious miniaturization system consists of a piezoelectric substrate and a fixed droplet placed directly on a piezoelectric platform [11]. It avoids the complicated fabrication and bonding process of the microfluidic channel, and requires no additional equipment, such as external pump sources and so on [12]. It can break through the limitation of liquid volume and realize reuse [13], especially in microliter microfluidic processing [14], and this method has been used in droplet spraying [15], droplet atomization [16], particle agglomeration [17], cell separation [18], cell cleavage [19] and so on.

In order to effectively analyze the forces of suspended particles in a sessile droplet, it is necessary to comprehensively analyze the propagation and attenuation of SAW in droplets. The aggregation of nano-diamond particles is realized by Asma Akther et al., and particle rings appear in the process of particle capture. They attributed the capture of the particles to centrifugal force and drag force caused by acoustic streaming flow (ASF), and ignored the influence of acoustic radiation force (ARF) [20,21]. The influence of ARF caused by leakage wave (Leaky-SAW) is considered by Ghulam Destgeer et al. in the study of particle agglomeration behavior. Although Destgeer successfully discovered and verified the existence of particle rings, the influence of signal intensity on acoustic streaming flow was ignored. The signal intensity determines the leakage of acoustic amplitude and energy in the liquid, which affects the vortex velocity and the centrifugal force of suspended particles [22]. Therefore, our study begins with ARF caused by Leaky-SAW, drag and centrifugal forces caused by ASF, and explores the differences in particle capture shapes caused by different dominant forces. In this paper, a novel method for realizing free transformation between clusters and rings of particles by modulating the acoustic amplitude of travelling SAW is proposed.

The movement of fluorescent particles in the sessile droplet was observed by fluorescence microscope, and the process of capture was recorded by a high-speed video. Then, center normalized pixel intensity analysis (NPI) was performed on the single frame image extracted from the video to determine the capture position and intensity. By adjusting the energy intensity of the excited acoustic surface waves to control the acoustic amplitude, the velocity of the vortex in the droplet can be regulated freely (see Figure 1). Then, it is observed that the suspended particles captured by the acoustic field-induced fluid vortices (acoustic vortices) are in the shape of clusters or rings (see Figure 4). Repeated experiments on Polystyrene (PS) fluorescent particles with diameters of 5, 20, and 40 μm at frequencies of 99.1, 48.8, and 20.4 MHz clearly proved this phenomenon (see Figure 5).

An ingenious microdevice was used to produce a unidirectional ASF at any position, without influence by the initial position of the droplet. This allows the particles to follow the vortex in a certain direction and speed, which plays an important role in particle agglomeration. In our further demonstration, a novel approach of free transition between the particle ring and cluster can be approached by modulated the energy intensity of SAW in our further demonstration, which provides the possibility of particle dilution and concentration at a certain concentration (see Figures 6 and 7). At the same time, due to the different SAW radiation intensity of particles with different particle sizes, the aggregation of particles in the droplet has a certain order, so that the formation of 20 μm wrapped 40 μm and 20 μm wrapped 5 μm particle clusters were obtained (see Figure 9).

## 2. Experimental Set-Up, Materials and Methods

### 2.1. Experimental Set-Up, Materials and Methods

To produce a steady flow of acoustic streaming in stationary droplets, which are usually placed on the edge of an inter-digital transducer (IDT) [23], or an oblique IDT is used [24] so that the acoustic energy is transmitted unevenly into the droplets, resulting in an acoustic streaming ASF (Eckart flow) [25]. However, both methods require precise control of the droplet position, which is challenging in operational practice and application scope is limited [17]. For comparison, we designed two simulation models: the first one is the symmetric propagation of acoustic waves and the second one is the asymmetric propagation of acoustic waves using a curved-edge piezoelectric substrate (the model of this study). Simulations show that our model can produce a unidirectional ASF at any position, without influence by the initial position of the droplet (see Figure 1). In contrast, a normal model requires adjustment of the droplet placement. This is well illustrated by numerical simulation of the propagation and radiation of TSAWs in a droplet (See Appendix A for details).

It has been proved that when the SAW attenuation length (*x_s_*) is greater than the droplet radii (*r_d_*), the leaky-SAW will reflect from the boundary of the droplet and interfere with the incident wave to form a standing wave field. This situation occurs especially when the frequency is lower than 20 MHz and the droplet radius is smaller than 5 μL, 1.5 mm [26,27]. Therefore, in order to ignore the influence of the low-frequency standing wave field on the particles, the droplet volume at 8 μL has been fixed in this work, and designed SAW devices with resonant frequency greater than 20 MHz.

The Interdigital Transducer (IDT) with Cr and Au (20 and 100 nm, respectively) was fabricated on 500 μm thick, 128°Y-cut, X-propagating and double-polished LiNbO_3_ substrates by magnetron sputtering and lift-off processes. The resonant frequencies of travelling SAW were measured by using Vector Network Analyzer (E5071C, Keysight, Pulau Pinang, Malaysia) and RF Probe Station (PW-600, ADVANCED, Taiwan, China). The operating frequencies of 99.1, 48.8, and 20.4 MHz are the frequencies we chose to test, which are in the operating frequency range of the SAW devices. The RF AC signals were modulated and amplified by using Vector Network Analyzer and RF power amplifier (8447F, HP, Bosque Farms, NM, USA), and the output power was controlled by adjustable stepping attenuator (KST-30, REBES, Suzhou, China). The particles used for testing are 5 μm (fluorescence excitation is green), 20 μm (fluorescence excitation is blue) and 40 μm (fluorescence excitation is orange) polystyrene fluorescence microspheres. The fluorescent particles were observed by fluorescence microscope (NIB620-FL, Nexcope, Ningbo, China), and the microscopic video was recorded by 6.3 million pixels high-speed CMOS camera (Nexcam-TC6CCD, Nexcope, Ningbo, China), which finally analyzed by the ImageView software.

### 2.2. Methods

As travelling SAW propagates into the fluid interior, the acoustic energy of SAW radiates at the Rayleigh angle (*Snell’s law: sin^−^*^1^*(c_f_/c_s_*)) inside the fluid [28], where *c_s_* ≈ 3950 ms^−1^ and *c_f_* ≈ 1480 ms^−1^ are the speeds of sound in piezoelectric (LiNbO_3_, lithium niobate) substrate and fluid (water), respectively. The lost energy is converted into mechanical and internal energy of the liquid, so that the liquid flows turn into vortex, which is known as acoustic streaming flow (ASF). The unlost energy continues to propagate through the liquid in the form of the Leaky-SAW. Thus, suspended particles in the droplet are controlled by two mains force under the action of SAW: The drag force caused by ASF and the ARF caused by Leaky-SAW (Figure 2).

In order to characterize and compare the size of ARF, Ref. [29] an acoustofluidic dimensionless parameter *κ(=ka)* is defined, where *k(=*2*π_f_/c_f_)* is the wavenumber, *f* is the frequency of waves, *c_f_* is the speed of sound in the fluid and *a(=d_p_/*2*)* is the radius of the particle. The κ-factor has previously been used to describe the separation [30] and enrichment [31] of PS particles by travelling surface acoustic waves (TSAWs). ASF-based drag force was reported to dominate the particle motion when *κ <* 1, and if *κ >* 1, the leaky SAW-based ARF dominates [29,32,33]. When PS particles suspended in water are exposed to a definite acoustic field, the acoustic radiation force (*F_ARF_*) exerted on the particles is only a function of the particle dimensions *d_p_* [34]. Therefore, the operating frequencies used in this study are theoretically predictable.

The displacement amplitude Â is an important parameter for this work, which only influences the energy of SAW in the investigated power range. The estimation of Â with the supplied electrical power $P_{el}$ is based on the following relationship [35].
(1)$$Â=\sqrt{\frac{k_{\eta }P_{el}\lambda_{SAW}}{AP\cdot F_{N}}}$$
where AP=2.3 mm is the aperture of the IDTs and $F_{N}=2.47\times 10^{15}\;W/m^{2}$ is a material dependent factor. $k_{\eta }\in \left[0,1\right]$ is an additionally implemented parameter, which represents the loss due to the reflection of the voltage signal and the propagation attenuation during the excitation of the acoustic surface wave. In our experiments, TSAW propagates without any obstruction before entering the droplet, so the propagation attenuation can be ignored. Here we assume that there is no the reflection of the voltage signal and take $k_{\eta }=1$. The correlation between acoustic amplitude Â and the supplied electrical power $P_{el}$ can be obtained in Figure 3.

In addition, the centrifugal force is also a non-negligible force which affects the movement of particles in vortex. The vortex velocity is related to the change in the amplitude of the acoustic wave as it enters the droplet, which determines the magnitude of the centrifugal force. It is well known that centrifugal force *F_C_* is proportional to the square of angular velocity of centrifugal motion *ω (F_C_*
*∝ ω*^2^*)*. The centrifugal force increases when particles spin along the vortex at high speed. It begins to dominate the motion of the particles when the centrifugal force is large enough to exceed the acoustic vortex drag force. In other words, due to the different dominant force in the droplet, the particles will be captured by TSAW-induced acoustic vortex and show different trapped forms. Three types of particle capture are proposed in this paper: particle streaming, particle cluster and particle ring. The process of trapping these particles is described in detail below.

#### 2.2.1. Particle Streaming and Particle Cluster

For contrast, the streaming and cluster of particles was tested simultaneously (see Figure 4a–c). The operating frequency of SAW device and RF power were set as 48.8 MHz and 10 dBm, respectively, and fixed the droplet volume as 8 μL for the rigor of experiments. The 5 μm (green) and 20 μm (blue) polystyrene fluorescent microspheres were used for the experimental tests in this work.

It has been reported that when *κ <* 1, the scattering effect of surface acoustic wave of particles can be approximated as an isotropic sphere. Given the particle diameter is small compared to the wavelength *λ_w_**(=c_f/_f)* of the SAW in the fluid, ARF cannot change the particle motion, and the effect is almost negligible [34]. When *d_p_* ≪ *λ_w_* (≈30.3 μm, 5 μm particles in this experiment), the scattering effect of SAW can be equivalent to spherical isotropy with almost no backscattering. This SAW-based ARF caused by leakage has minimal effect on the particles and cannot effectively control the movement of 5 μm particles (see Figure 2a). In addition, due to the small velocity of the vortex with 10 dBm power applied, the centrifugal force of the particles is weak. The drag force caused by ASF plays a dominant role (top and side views in Figure 2b) and drags 5 μm particles along the acoustic vortex streamlines eventually (see Figure 2c), which is called Particle Streaming. The complete movement process is shown as the green fluorescent particles in Figure 4a–c.

The particle diameter *d_p_* (=20 μm) cannot be neglected compared to the wavelength *λ_w_* (30.3 μm ≈ *c_f/_f*) in the fluid when *κ >* 1 for the 20 μm particles in this experiment, and the scattering is no longer isotropic and backscattering dominates [34]. At this point, the ARF caused by Leaky-SAW would change the trajectory of the particles (see Figure 2d). In addition, the applied power (10 dBm) is small and the centrifugal force is difficult to handle, the leaky SAW-based ARF will play a dominant role *(F_C_ < F_ASF_ < F_ARF_)* to control the particles ultimately. Thus, the leaky SAW-based ARF changes the trajectory of the particles when the 20 μm particles are flowing through the Leaky-SAW region, and pushes them from the outer vortex line to inner line (see Figure 2e). Eventually, the particles are pushed to the center of the vortex, where they gather into a particle cluster (see Figure 2f). The complete moving process is shown as the blue fluorescent particles in Figure 4a–c, which is named as Particle Cluster in this paper.

#### 2.2.2. Particle Ring

For the experiment process of particle ring formation, the intensity of the applied signal (RF power) has been changed only, which was carried out by using 20 μm (blue) polystyrene fluorescent microspheres (see Figure 4d–f) and other conditions consistent with the previous tests (48.8 MHz, 8 μL, 20 μm). The particle ring of the 5 μm particles was difficult to observe due to the parameter limitation of *κ <<* 1.

As mentioned previously, the 20 μm particles will passing through the outer-side of the acoustic flow vortex to the inner streamline, and gradually approach the center when *κ* > 1, which is driven by the ARF. The acoustic amplitude attenuates rapidly within the fluid and more loss of energy happens as the signal intensity increases to 30 dBm. The wasted energy is converted into mechanical energy of the fluid, which is driving a faster vortex motion. Therefore, the centrifugal force (*F_C_*) exerted on the particles in the fluid gradually increases and becomes dominant (see Figure 2g,h). As shown in Figure 4e, the 20 μm particles moved along the streamline of the ASF and toward the center of the acoustic vortex initially. Then the enhanced centrifugal force pushed the 20 μm particles away from the vortex center when rotation speed of the vortex reached its maximum, as the ARF is unable to confine them. The 20 μm particles moved from inner line to outer line of vortex to form a stable particle ring when *F_C_ ≈ ARF* and reached an equilibrium state (see Figure 2i). The ARF pushes the particles towards the center of acoustic vortex, while the enhanced centrifugal force pushes the particles away from the center, which is opposite to the ARF. Therefore, the *F_C_* and the ARF jointly dominate the ring motion of particles around the streamlines of a specific acoustic vortex, which is named as the Particle Ring. The whole movement process is shown as the blue fluorescent particles in Figure 4d–f. The difference of 20 μm fluorescence color is caused by the different intensity of excitation light source.

Combined with the experiments described previously, under the condition of signal intensity of 48.8 MHz and droplet volume of 8 μL, it is observed that 5 μm particles move along the streamline of acoustic vortex, and 20 μm particles are pulled to the center of acoustic vortex to gather into a cluster. The 20 μm particles break away from the center of acoustic vortex and are concentrated to a particle ring when the signal intensity increased to 30 dBm. (See Appendix A for more details) We reasonably explained these phenomena by the different dominant effects of three forces: drag force caused by ASF, ARF caused by Leaky-SAW and centrifugal force. It is concluded that the suspended particles can be captured by the acoustic field-induced fluid vortex (acoustic vortex), whose shape is particle cluster or particle ring.

## 3. Results

### 3.1. Testing of Particle Ring and Cluster with Different Particle Diameters

In order to verify the reproducibility of the experimental results, formations of 5 μm (green), 20 μm (blue) and 40 μm (orange) PS fluorescent particles of the particle cluster or ring were tested additionally (see Figure 5).

Under the influence of the acoustic surface wave field with the operating frequency of 99.1 MHz, 5 μm particles clustered at the signal intensity of 10 dBm (see Figure 5d), and then formed a stable particle ring when the signal intensity increased to 30 dBm (see Figure 5a). The was similar to the 20 and 40 μm particles, which clustered at *f_SAW_* = 48.8 MHz and RF power = 10 dBm (see Figure 5e), and then formed a stable particle ring when the signal intensity increased to 30 dBm (see Figure 5b,c,f). The results show that the aggregation behavior of small particles can be achieved by high-frequency acoustic waves, because smaller wavelengths in the fluid can have a stronger backscattering effect on small particles, which also validates the previous results.

ImageView software was used to measure the normalized fluorescence intensity of each column in Figure 5. The vertical axis represents the normalized intensity value and the horizontal axis represents the normalized distance. It is easy to identify from the normalized curves (see Figure 5h–g) that the particles with ring form (black curve) are mainly distributed at the outer ring of the vortex, while the cluster particles (green, blue and orange curves) are almost concentrated at the center of the vortex.

### 3.2. The Free Transition of Particle Ring and Particle Cluster

Previous Particle Ring experiments show that with the increase TSAW acoustic amplitude, particle clusters gradually form particle rings. In order to prove this phenomenon works the other way as well (particle ring also shrinks into particle cluster when the applied TSAW acoustic amplitude decreases), an experiment shown in Figure 6a–f had been carried out for this purpose. It can be seen that the acoustic amplitude gradually decreases as the acoustic power decreases, which leads to a decrease in the vortex velocity. Then, the diameter of the particle ring gradually decreases until it shrinks into a particle cluster. (See Appendix A for details). According to the normalized pixel intensity (NPI) value measured from the vortex center (Figure 6g), the change of NPI value at different input power can be reflected by the coverage area of different colors. Characterization of these color coverages indicates that the concentration distribution of particles in the droplet can be controlled by TSAW-induced acoustic vortex. The diagram of the particle outer/inner ring with the signal intensity configurates the shrinking process more visually (Figure 7). The decrease of the outer and inner diameter of the particle ring is consistent with the decreasing power intensity.

To further quantify the effect of acoustic amplitude on particle rings, the acoustic amplitudes Â at different RF power (dBm) have been calculated as shown in Table 1.

It is obvious that the amplitude of the acoustic displacement in the substrate and the droplet is different and increases with the increase of the electrical power. The correlation between the inner and outer diameters of the particle ring and the square of the acoustic amplitude Â is shown in Figure 8.

The energy magnitude of the acoustic leakage determines the size of the particle ring morphology in the droplet. It can be seen that in the first half of the increase of the acoustic amplitude (0–5 Å), the inner and outer diameter of the ring changes more significantly. As the acoustic amplitude continues to increase (5–15 Å), the ring increases more slowly and gradually stabilizes, indicating that a more stable control of the ring morphology can be achieved at larger acoustic amplitudes, which is consistent with our test Appendix A. Therefore, a conclusion can be generalized that the conversion between particle clusters and particle rings can be realized freely by adjusting TSAW acoustic amplitude, which proposes a possibility to dilute and concentrate particles to a certain concentration.

## 4. Discussion

In the shrinkage test of particle ring, particles with different size shrink into particle clusters in a specific order. As can be seen from Figure 9a, under the control of acoustic vortex induced by TSAW (20.4 MHz), the 40 μm (orange) particles shrink to the vortex center earlier than 20 μm (blue) particles, and then form a multilayer particle cluster structure. Figure 9c demonstrates the formation of a 20 μm multilayer particle mass around 5 μm under the control of a (99.1 MHz) TSAW-induced acoustic vortex. The values of NFI plot in Figure 9e,f illustrate that most 40 μm (orange) particles and 5 μm (green) particles are gathered in the center of the multilayer particle aggregates, while 20 μm particles are distributed at the periphery of the aggregates (see Figure 9b,d).

The reason for this phenomenon may due to the domination of ARF in driving the particles as the centrifugal force has been decreased. Destgeer et al. [29] demonstrated the acoustic radiation force is maximum at κ ≈ 1.4 [36] after the derivation of the formula κ = πdpf/cf. Compared to 20 μm particles, the κ factors of 40 and 5 μm particles are closer to 1.4, which is represented in the present experiment (κ40 μm ≈ 1.73, κ20 μm ≈ 0.87, fSAW = 20.4 MHz) (κ5 μm ≈ 1.05, κ20 μm ≈ 4.2, fSAW = 99.1 MHz). Based on this, the particles are subjected to a larger ARF in the acoustic field, which leads to the formation of multilayer particle aggregates with 20 μm covering 40 μm and 20 μm covering 5 μm, before the 20 μm concentrated to the vortex center.

## 5. Conclusions

Suspended PS particles with different sizes (5, 20, 40 μm) in a sessile droplet have been captured successfully by using three TSAW-based devices with different frequencies (99.1, 48.8, 20.4 MHz). We proposed a novel method of free transition between particle clusters and rings by modulating the travelling of SAW acoustic amplitude. The smaller particles (with *κ* < 1) move along the acoustic vortex streamlines, while the larger particles (with *κ* > 1) are pulled toward the center of acoustic vortex to gather into a cluster, or escape from the center of the acoustic vortex to gather into a particle ring. These phenomena can be explained by the domination effects of three different forces, which are drag force caused by ASF, ARF caused by Leaky-SAW and varying centrifugal force. It is concluded that the acoustic field-induced fluid vortex (acoustic vortex) can capture suspended particles in the shape of particle clusters or particle rings. Subsequently, by adjusting the acoustic amplitude of TSAW to modify the intensity of acoustic vortex effect, the size of the particle ring can be controlled freely, and the multilayer particle aggregates with 20 μm wrapped around 40 μm and 20 μm wrapped around 5 μm are obtained. In the future, it is expected to open a new way for precise concentration and quantitative analysis of particles or cells at low concentration.

## Figures and Tables

**Figure 1 biosensors-12-00399-f001:**
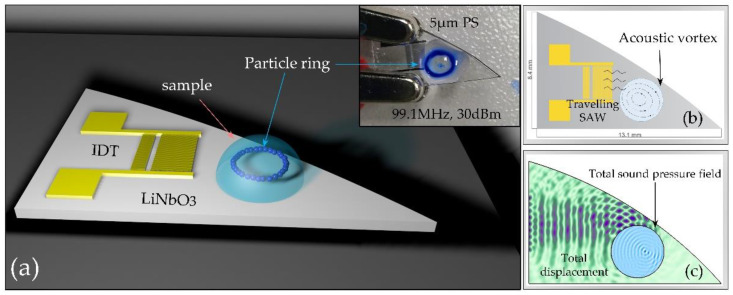
(**a**) Schematic diagram of the acoustofluidic device: Curved-edge cut lithium niobate substrate and sample droplets placed on the surface. The inset depicts the particle ring shape was exhibited under acoustic vortex capture with 5 μm particles. (**b**) Design diagram of the device. Acoustic vortex induced by asymmetric propagation of Travelling SAW excited by the interdigitated transducer (IDT) in a sample droplet. (**c**) Numerical simulation of the propagation and radiation of TSAWs in water.

**Figure 2 biosensors-12-00399-f002:**
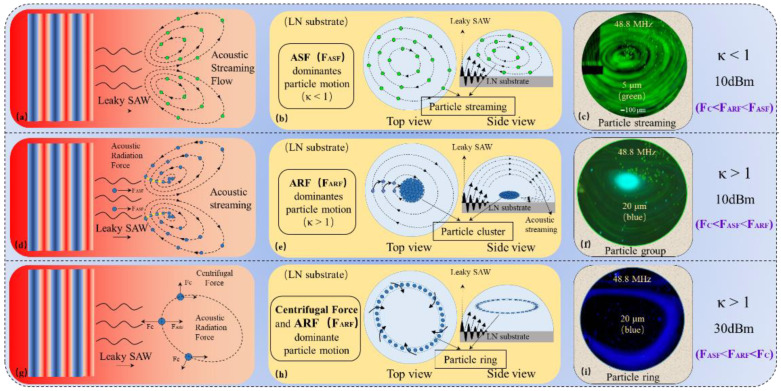
(**a**–**c**) Demonstrate that when ASF > ARF (κ < 1, 10 dBm), the drag force caused by ASF dominates the vortex motion of the particles (5μm) along the acoustic vortex streamlines.—Particle Streaming. (**d**–**f**) Demonstrate that when ASF < ARF (κ > 1, 10 dBm), the ARF caused by Leaky-SAW dominates the motion of the particles (20 μm) towards the center of the acoustic vortex. —Particle Cluster. (**g**–**i**) Demonstrate that when ASF < ARF (κ > 1, 30 dBm), the ARF caused by Leaky-SAW pushes the particles (20 μm) towards the acoustic vortex center, but the enhanced centrifugal force drives the particles out of the center again and against the ARF. The *F_C_* and ARF co-dominate the ring motion of the particles around a specific acoustic vortex streamline.—Particle Ring.

**Figure 3 biosensors-12-00399-f003:**
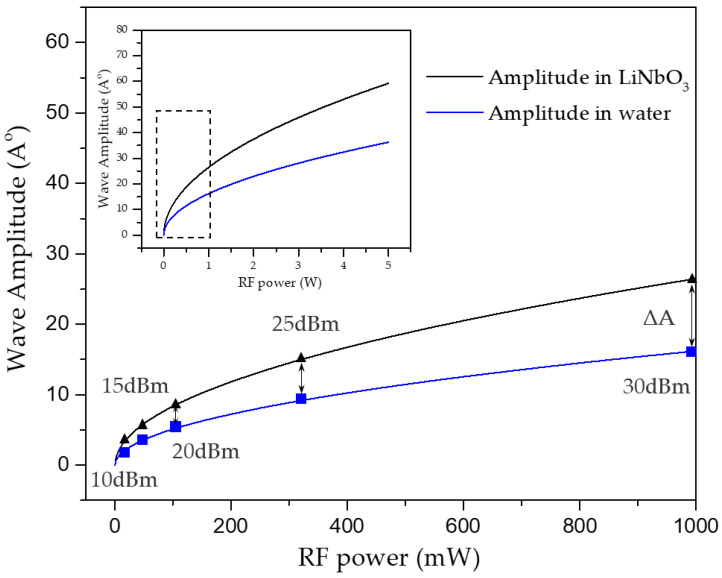
Acoustic displacement amplitudes Â at different electrical power levels inside the piezoelectric substrate and droplet, respectively. The frequency of the excitation acoustic wave is constant at 99.1 MHz.

**Figure 4 biosensors-12-00399-f004:**
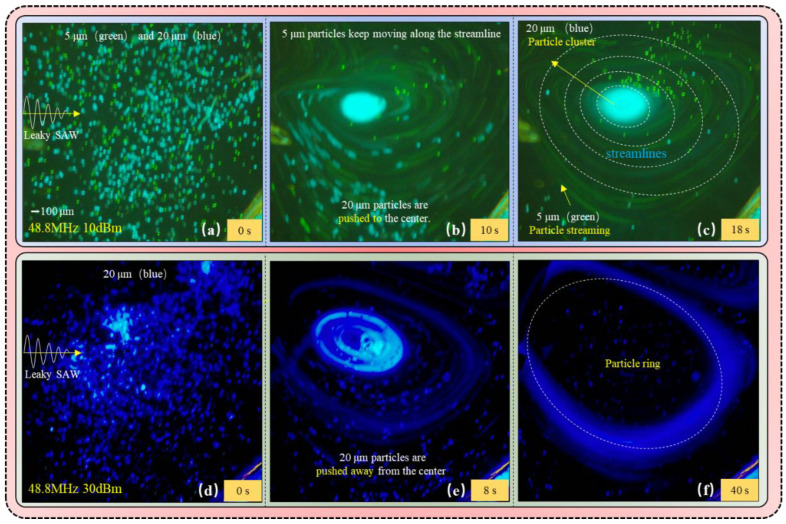
(**a**–**c**) Describes the formation process of 5 μm Particle Streaming and 20 μm Particle Cluster at 48.8 MHz, 10 dBm. The 5 μm particles tend to move along the streamlines of the ASF, but not across over to the other streamlines, forming the particle streaming. The larger 20 μm particles are driven by the ARF to move along the streamlines of the ASF from the outer streamline to the inner streamline, and gradually concentrate in the center of acoustic vortex. (**d**–**f**) Describes the formation process of 20 μm Particle Ring at 48.8 MHz, 10 dBm. The ARF pushes the particles to the center of acoustic vortex, but the enhanced centrifugal force pushes the particles away from the center of acoustic vortex. The *F_C_* and ARF co-dominate the ring motion of the particles.

**Figure 5 biosensors-12-00399-f005:**
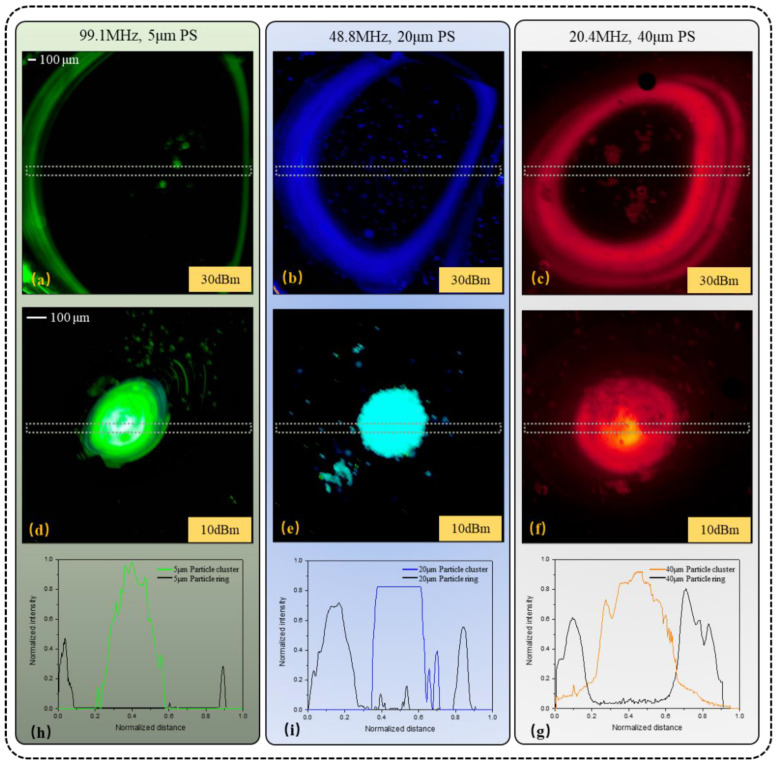
Summary of the experimental manipulation of different diameter (5, 20, and 40 μm) polystyrene (PS) particles suspended in DI water using TSAWs of different frequencies (90.1, 48.8, and 20.4 MHz). Screenshots of the sessile droplets loaded with particles were captured after the SAW was exposed at 30 dBm (**a**–**c**) and 10 dBm (**d**–**f**) RF power for periods sufficiently long to attain the desired particle concentrations at the target locations. The normalized fluorescence intensities (NFI) values measured from the droplet centers (see the dashed rectangles in (**a**–**f**), from left to right) are plotted in (**h**–**g**) to characterize the migration characteristics of the particles at different signal intensities.

**Figure 6 biosensors-12-00399-f006:**
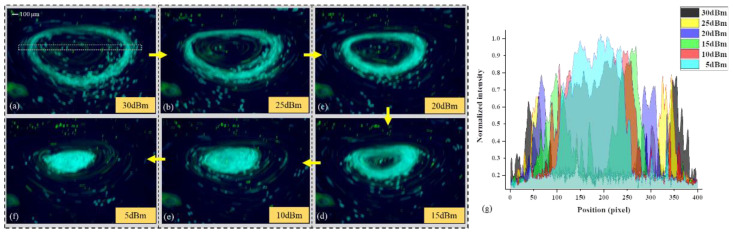
(**a**–**f**) Describes the shrinking process of particle ring with different input powers. The normalized pixel intensity (NPI) values measured from the center of the vortex (see the dashed rectangles in (**a**), from left to right) are plotted in (**g**) to characterize the concentration of particles at different signal intensities.

**Figure 7 biosensors-12-00399-f007:**
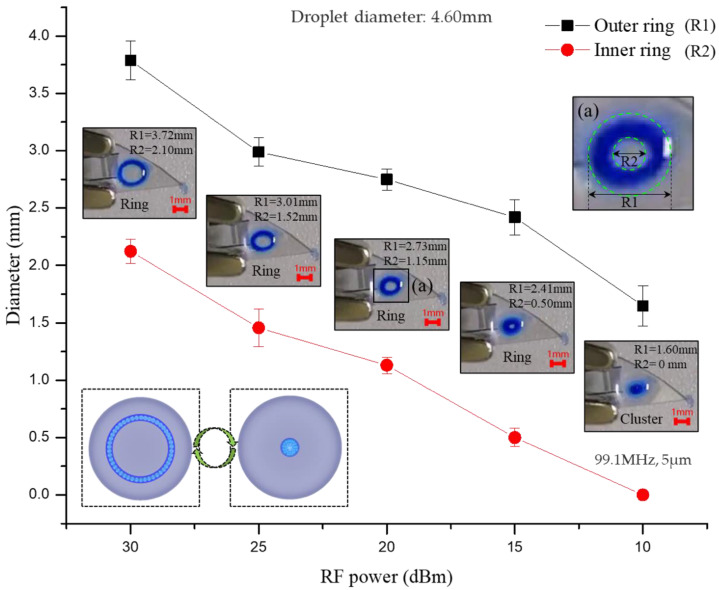
Graph of the variation of the outer/inner ring diameter with the input power during the shrinkage of the particle ring.

**Figure 8 biosensors-12-00399-f008:**
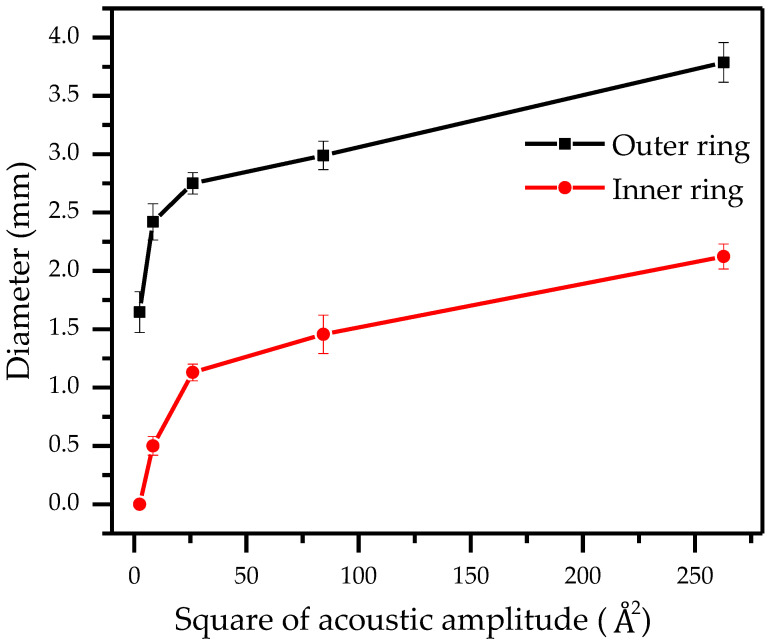
The correlation between the inner and outer diameters of the particle ring and the square of the acoustic amplitude.

**Figure 9 biosensors-12-00399-f009:**
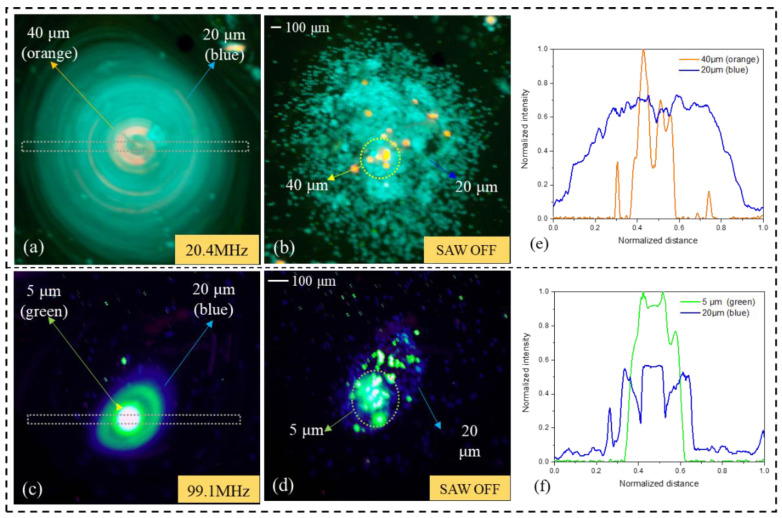
Multilayered particle agglomerates of 20 μm wrapped around 40 μm (**a**,**b**) and 20 μm wrapped around 5 μm (**c**,**d**). The normalized fluorescence intensities (NFI) values measured from the vortex centers (see the dashed rectangles in (**a**,**c**), from left to right) are plotted in (**e**,**f**) to characterize the distribution of particles of different sizes.

**Table 1 biosensors-12-00399-t001:** Acoustic displacement amplitudes Â in the substrate and droplet.

RF Power (dBm)	10	15	20	25	30
Amplitude in LiNbO_3_ (Å)	2.51	4.71	8.34	14.90	26.49
Amplitude in water (Å)	1.54	2.88	5.10	9.18	16.21

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
