# Peer review of "Enhanced Detection in Droplet Microfluidics by Acoustic Vortex Modulation of Particle Rings and Particle Clusters via Asymmetric Propagation of Surface Acoustic Waves"

_biosensors, 2022, doi:10.3390/bios12060399_

Round 1

Reviewer 1 Report

The authors presented a simple active technology that uses a programmable acoustic vortex created by surface acoustic fields to accurately concentrate biological materials such as particles or cells at low concentration. The experimental setup and data collection process are well-designed, and the gathered results are evaluated and presented correctly. After some minor changes, it appears to be acceptable.

(1) In the introduction, describe what makes it unique (or superior) to previous studies that control particle behaviors by excitation of bulk or surface acoustic fields (e.g., Sensors and Actuators B: Chemical., Vol. 246, pp. 415, 2017).

(2) It is necessary to explain whether the resonant frequencies used in this study are theoretically predictable.

(3) Please explain how the quantity or arrangement of lithium niobate substrates that generate SAW affects particle concentration.

Reviewer 2 Report

This study uses MHz surface acoustic waves for the control of micro-particules clustering via acoustic radiation force combined with acoustic streaming. Although the experiments are carefully designed, there is a lack of originality in this study since this phenomenon was observed in various previous studies.

A more thorough and quantitative study would be needed varying acoustic power, measured by the direct determination of the acoustic displacement of the substrate, and not by electric power since this is very much system-dependent. Laser interferometry is often used on this purpose.

The influence of frequency is not sufficiently worked through, and a clear argument should be given in terms of which is the dominant effect between acoustic streaming and radiation force, also considering particle size.

Reviewer 3 Report

Liu et al. presented Enhanced detection in droplet microfluidics by acoustic vortex modulation of particle rings and particle clusters via asymmetric propagation of surface acoustic waves. In this work, travelling surface acoustic weaves (TSAWs) are applied to form micro particle aggregation and rings by the effects of acoustic radiation forces, acoustic streaming and Stokes drag force within a droplet. By modulating the characteristics of the TSAWs, particles are changed from cluster aggregation to ring formation. This method is presented as a new approach to on-demand dilute or concentrate particles in a droplet.  Overall, the manuscript is well-written. The data presented in the paper supports the claims of the authors. Especially Figures 6 is a good summary of the method as a visual demonstration. I have the following specific comments.

*In general font size in the figures should be increased. It is difficult to read the smaller texts in the figures

*An estimation of the value of the ARF could be useful. 

*No error bar is shown in Fig 6 for the ring diameters. How consistent is the change of the ring diameter with the decreasing power intensity. 

Round 2

Reviewer 2 Report

The corrections made in this new versions are minimal. I acknowledge the effort made by the authors to clarify a few points, but my initial concern remains. Therefore, the results obtained remain qualitative and of limited impact, since there is no measurement of the acoustic amplitude.
